# Evaluation of the Wear and Mechanical Properties of Titanium Diboride-Reinforced Titanium Matrix Composites Prepared by Spark Plasma Sintering

**DOI:** 10.3390/ma16052078

**Published:** 2023-03-03

**Authors:** Olusoji Oluremi Ayodele, Bukola Joseph Babalola, Peter Apata Olubambi

**Affiliations:** Centre for Nanoengineering and Tribocorrosion, School of Mining, Metallurgy and Chemical Engineering, University of Johannesburg, Johannesburg 2028, South Africa

**Keywords:** ceramic reinforcement, microstructures, nanomechanical properties, tensile behavior, titanium alloy, tribology

## Abstract

The synthesis of x-wt.% (where x = 2.5, 5, 7.5, and 10) TiB_2_-reinforced titanium matrix was accomplished through the spark plasma sintering technique (SPS). The sintered bulk samples were characterized, and their mechanical properties were evaluated. Near full density was attained, with the sintered sample having the least relative density of 97.5%. This indicates that the SPS process aids good sinterability. The Vickers hardness of the consolidated samples improved from 188.1 HV1 to 304.8 HV1, attributed to the high hardness of the TiB_2_. The tensile strength and elongation of the sintered samples decreased with increasing TiB_2_ content. The nano hardness and reduced elastic modulus of the consolidated samples were upgraded due to the addition of TiB_2_, with the Ti-7.5 wt.% TiB_2_ sample showing the maximum values of 9841 MPa and 188 GPa, respectively. The microstructures display the dispersion of whiskers and in-situ particles, and the X-ray diffraction analysis (XRD) showed new phases. Furthermore, the presence of TiB_2_ particles in the composites enhanced better wear resistance than the unreinforced Ti sample. Due to dimples and large cracks, ductile and brittle fracture behavior was noticed in the sintered composites.

## 1. Introduction

Titanium and its alloys (Ti) have an outstanding combination of high strength, high elastic modulus, low density, good biocompatibility, and corrosion resistance [1,2]. On this account, Ti has become a well-known material for various technological and industrial applications in the biomedical, transportation (aerospace), and chemical sectors [3,4,5]. However, its industrial applications have been restricted due to low wear resistance and stiffness [6]. To overcome such limitations, and for Ti to compete with superalloys, titanium composite approaches have been developed. Ti-based composites have been utilized in aerospace and automotive applications because of their unique mechanical properties, wear, and corrosion properties [6,7,8]. Some researchers have reported that ceramic reinforcements can improve the mechanical attributes of titanium composites [9,10,11].

Ceramic reinforcements like TiB_2_, SiC, TiC, and TiN possess excellent wear resistance, high hardness, and thermal stability [12]. On the other hand, TiB_2_ is among the ultra-high-temperature ceramics consisting of excellent elastic modulus, high melting point, and good electrical conductivity [13]. TiB_2_ has been incorporated with Ti-based composites to improve their mechanical properties, according to Ayodele et al. [14]. They investigated the effect of TiB_2_ reinforcements in the SPS of Ti composites. The outcome improved relative density and microhardness values by an average of 3.4% and 28.5%, respectively. Singh et al. [15] also showed that the effect of TiB_2_ contents (2 wt.%, 5 wt.%, 10 wt.%, and 20 wt.%) improved the developed Ti composite’s flexural strength, wear properties, and impact results. However, it is worth noting that good bonding between the ceramics and matrix, as well as good consolidation routes, are necessary to achieve excellent mechanical properties in the composites [16].

In addition, one of the most widely consolidation routes utilized by many researchers includes the powder metallurgy route (PM). PM is widely known for fabricating complex and near-net-shaped materials by blending alloyed or unalloyed powders [16,17]. The benefit of PM over the casting method and combustion synthesis includes its cost-effectiveness in densifying particulate reinforced metal matrix composites [18,19]. Lately, a PM route that has received broad attention is the spark plasma sintering technique. Spark plasma sintering (SPS), also known as pulsed electric current sintering (PECS), can consolidate near-fully dense samples at a short sintering cycle and lower temperatures [20]. Thus, this leads to good mechanical properties and uniform microstructures compared to traditional sintering methods [21,22]. In this technique, alloyed powders are mechanically pressed into the sintering die, and a pulsed direct current is applied to fabricate the powders [23]. SPS accelerates the transfer of material and diffusion due to plasma formation within the particles. Thus, impurities are removed from the sample surface, and the sinterability of the sample is aided by the joule effect and pressure-assisted plastic deformation [24,25]. Furthermore, SPS can consolidate different materials, including ceramics, metals, composite materials, etc. [26,27,28,29,30].

Previously, Mohammadzadeh et al. [31] investigated the effect of sintering temperature from 750 °C to 1350 °C on the microstructure and densification of Ti-4.8 wt.% TiB_2_ composite using the SPS technique. The relative density of the fabricated composite increased from 93.9% to 99.8% with the sintering temperature, with the sintered composite at 1200 °C indicating the highest relative density. The microstructures observed were a needle-like phase of TiB as the temperatures were above 900 °C. Ti-4.8 wt.% TiB_2_ composite at 1200 °C had the best tensile strength value (541 MPa) and elongation (6.62%) among other composites by virtue of their high relative density and high-volume fraction of TiB phases. In another study by Namini et al. [32], the influence of in-situ-formed TiB phases was observed on the spark plasma sintering of 2.5 wt.% TiB_2_/Ti and 1 wt.% B4C/Ti composites. The study shows the formation of TiB, TiC, and Ti in the XRD spectrum of both composites. Furthermore, the relative density of the titanium matrix improved upon adding both composites by over 0.6% increments. Both composites exhibited a 10% increment in hardness and tensile strength and a 30% decrease in bending strength than the sintered titanium matrix.

These investigations have successfully exploited the properties of TiB_2_ in improving the microstructures and mechanical properties of the Ti matrix. However, the wear properties of the Ti-TiB_2_ composite have yet to be sufficiently reported in the literature. Hence, this work aims to fabricate a Ti matrix with the variation of TiB_2_ content from 2.5 wt.% to 10 wt.% via the novel sintering approach and to examine the distinct phases, microstructures, wear properties, fractured surfaces, and the mechanical properties of Ti-TiB_2_ composites.

## 2. Experimental Procedures

### 2.1. Materials 

Ti (25 µm particle size, 99.8% purity, TLS Technik GmbH & Co., Bitterfeld, Germany) and TiB_2_ (8–41 µm particle size, 99.9% purity, WearTech Ltd., Port Talbot, UK) were utilized as the starting raw materials.

### 2.2. Morphological Characterization

The powders were placed in a plastic vessel and loaded in a Turbula machine for 7 h. The admixed samples were placed in a graphite die and consolidated in a vacuum using the spark plasma sintering (Effelder-Rauenstein, Germany, HHPD-25). The fabrication procedure was performed at 1050 °C and 100 °C/min for 5 min. The compacts were sandblasted to remove undesirable graphite from the sintered samples. The sintered compacts were ground with Silicon carbide foil and papers of 5-disc sizes, and a 1-micron diamond suspension was used to polish the sintered samples. The sintered samples were etched with the solution comprising 5 mL of hydrofluoric acid, 10 mL of nitric acid, and 85 mL of distilled water to reveal the microstructures. X-ray diffractometer (Rigaku) was used to assess the distinct phases of the sintered samples using Cu–Ka radiation. A 1.5 deg/min scan speed was used over the 10° and 90° scan range. A voltage of 40 kV was used during the test. The crystallite size and lattice strain of the sintered samples were determined by the Williamson–Hall method. JSM-7900F and JEOL (SEM, scanning electron microscope) equipped with energy dispersive spectroscopy (EDS) was used to determine the morphology of the sintered samples. OLYMPUS GX5 was utilized to assess the optical images of the fabricated samples. The sample codes and graphical experimental process of the sintered samples are shown in Table 1 and Figure 1.

### 2.3. Mechanical Analysis

ZwickRoell Indentec was utilized to determine the Vickers microhardness of the consolidated samples at a load of HV1, a dwell time of 15 s, and an average of eight measurements recorded. The relative densities of the sintered sample were determined by the Archimedes principle. The experimental densities were recorded, and an average of six measurements were taken. A rule of mixtures was used to evaluate the relative density in relation to the theoretical density. An electrical discharge machining (EDM) was used to prepare the tensile samples using the dimensions (10 mm × 5 mm × 30 mm). An Instron machine (1195) was used to pull the test sample at a 4 mm/min strain rate using a load cell of 10 KN. The tensile data were used to plot the stress and strain curves.

### 2.4. Nanoindentation Test

The nanomechanical characteristics of the sintered samples were examined using a nanoindentation tester (Hit 300, Anton Paar, Graz, Austria). It consists of a diamond Berkovich indenter and a laser targeting the flat polished sample with an accuracy of <1 mm. The test was conducted according to ASTM E254 [33]. Furthermore, the samples were subjected to a load of 25 mN at a pause time of 10 s to ascertain the time-dependent deformation characteristics. Seven indentations were recorded for each sample. The nano mechanical properties were determined from the test (load-displacement data) in line with the Oliver and Pharr method [34]. The sample’s elastic modulus and hardness are the two nanomechanical properties measured frequently from the load-displacement curve. Hardness is defined as the ratio of the indentation load to the projected contact area, which is given by:(1)$$H=\frac{Maximum\;load\;\left(Pmax\right)}{Contact\;Area\;\left(A\right)}$$

Elastic modulus of sample indented can be inferred from the initial unloading contact stiffness; $S=\frac{dP}{dh}$, according to the relationship proposed by Sneddon [35] between the contact area, stiffness, and elastic modulus, which is represented by:(2)$$S=2\alpha \;\sqrt{\frac{A}{\pi }}*E_{r}$$

From Equation (2), α=1.034 for the Berkovich indenter. It is a constant depending on the geometry of the indenter [34]. $E_{r}$ is the reduced elastic modulus, which explains that elastic deformation occurs in both the indenter and the sample. It is expressed as:(3)$$\frac{1}{E_{r}}=\frac{1-V^{2}}{E}+\frac{1-Vi^{2}}{E_{i}}$$

V is the Poisson’s ratio, E is the elastic modulus, and $V_{i}$ and $E_{i}$ are the quantities for the indenter, respectively. For instance, the $V_{i}$ and $E_{i}$ of a diamond are 0.07 and 1141 GPa [33,34], respectively.

### 2.5. Tribology Study

The wear experiment of the sintered T0.0 samples and composites (T2.5, T5.0, T7.5, T10, respectively) was conducted using a ball-on-disk tribometer (Anton Paar, Graz, Austria). All the tests were carried out under dry conditions at 20 °C and a relative humidity of 50%. An Alumina ball (6 mm diameter) slid upon the surface of the sintered samples at a speed of 200 rpm. The test was performed at varied loads of 9 N and 13 N, respectively, with the linear speed of 5.17 cm/s and 11.41 cm/s. The friction coefficient was determined, and a profilometer (Taylor Hobson, Leicester, UK) was used to measure the depth and wear rate of the sintered samples. The test was carried out thrice to ensure repeatability.

## 3. Results and Discussion

### 3.1. Characterization of the Powders

The SEM micrographs and X-ray diffractograms of the starting raw materials were presented in our previous study [14]. The image of Ti powder showed a spherical satellite structure, signifying that the powder was produced by the gas atomization method [36,37]. The image of TiB_2_ powder displayed an uneven structure. In the XRD pattern of Ti powder, an αTi phase could be seen. Meanwhile, TiB_2_ powder showed the crystalline structure of the TiB_2_ phase, according to the X-ray diffractogram.

### 3.2. Evaluation of the Relative Density

Figure 2 shows the relative densities of the sintered samples. The T0.0 sample achieved a relative density of 97.5%. Meanwhile, T2.5, T5.0, T7.5, and T10 samples had relative densities of 97.9%, 99.2%, 99.3, and 99.1%, respectively. The addition of TiB_2_ reinforcement raised the relative densities of the sintered composites. The relative densities of the sintered composites increased as the TiB_2_ reinforcements increased, except for the T10 sample, which showed a slight decline in relative density, attributable to the high amount of TiB_2_ reinforcement caused by the nucleation and development of the in-situ phase formed in the composite that influences the densification procedure [14,38]. All the sintered composites achieved near-full densification, which proved that the SPS enables good sinterability.

### 3.3. Diffractograms of the Sintered Samples

Figure 3 shows the XRD spectra of the consolidated samples. The diffractograms of the T0.0 sample detected an alpha-Ti (DB card No: 03-065-9622), beta-Ti (DB card No: 00-044-1288), and Ti (DB card No: 00-044-1294) phases, respectively. The peak of the alpha-Ti and Ti phases were observed at 2θ = 35.06° 38.34°, 40.13°, 52.93°, 62.90°, 70.54°, 76.12°, and 77.29°, matching the planes (100), (002), (101), (102), (110), (103), (112), and (201). Meanwhile, the peak intensity of the beta-Ti phase was observed at 2θ = 38.48°, matching the plane (110). At 2θ = 26.3°, no phase was observed in the T0.0 sample. In the XRD spectrum of Sample T2.5, alpha-Ti (αTi, DB card No: 01-089-5009), titanium boride (TiB, DB card No: 01-073-2148), Boron titanium (TiB55, DB card No: 00-052-0845), and gamma-TiB_12_ (DB card No: 00-052-0843) phases were detected in the spectrum. The TiB_55_ phase had the highest peak intensity at 2θ = 26.54°, corresponding to the plane (300). The XRD spectrum of sample T5.0 revealed Ti (DB card No: 03-065-3362) and TiB (DB card No: 01-073-2148) phases, while the XRD spectrum of sample T7.5 revealed Ti (DB card No: 00-005-0682), TiB (DB card No: 01-073-2148), and TiB_55_ (DB card No: 00-052-0845) phases. In the XRD spectrum of sample T10, αTi (DB card No: 01-089-5009), TiB_55_ (DB card No: 00-052-0845), and Ti_2_B_5_ (DB card No: 00-006-0528) phases were detected in the spectrum. The peak intensities of the sample decreased supposedly due to more addition of reinforcement, resulting in reduced crystallite size [39]. TiB_2_ was not formed in the XRD pattern because the reaction between Ti and TiB_2_ was completed during the sintering process. Singh et al. [15] reported a similar investigation on the effect of TiB_2_ content on the microstructure and properties of Ti-TiB composites. However, the TiB_55_ phase showed the highest peak intensity at 2θ = 26.54°, corresponding to the plane (116).

In Figure 4, the XRD technique was used to analyze the crystallite size and lattice strain of the consolidated samples using the Williamson–Hall method. The consolidated T0.0 sample had crystallite size and lattice strain values of 135.7 Å and 0.124%, respectively. Meanwhile, the consolidated T2.5, T5.0, T7.5, and T10 samples had crystallite size values of 183.4 Å, 175.1 Å, 147.7 Å, and 137.5 Å, respectively. In addition, the lattice strain values were observed to be 0.205%, 0.242%, 0.216%, and 0.235%. The crystallite size values of the consolidated composites decreased with increasing TiB2 content. This implies that the strength of the composites would increase with increasing TiB_2_ content, according to the Hall–Petch equation [40]. In addition, the lattice strain values of the consolidated samples fluctuated with increasing TiB_2_ content, and high lattice strain values were recorded for T5.0 and T10 composites. However, an increase in the lattice strain may be due to an increase in the grain boundary fraction, mechanical deformation, and size mismatch effect between the constituents [41,42].

### 3.4. Microstructural Studies

#### Optical and SEM Microstructures

The optical images of the consolidated samples are shown in Figure 5. The T0.0 sample shows Ti’s α and β lamellar phases (Figure 5a). The α phase is light grey, while the β phase is dark grey. It is evident that when Ti is processed above 882 °C, the α phase transforms to the β phase during SPS processing. However, upon a fast cooling rate, the β phase remained at α phase grain boundaries because the reverse transformation was not complete [32]. The backscattered electron image of the sintered T0.0 sample is shown in Figure 6a, with the α and β lamellar structures. The microstructure of the sintered sample has a few pores indicating that full densification was not attained during SPS processing. Meanwhile, the optical images of the sintered TiB2–Ti composites are presented in Figure 5b,c. The TiB_2_–Ti composites (T2.5, T5.0, T7.5, and T10) showed the dispersion of the ceramic particles in the metal matrix as pointed with white arrows, while the red arrows represent the whiskers formed in the composites. The backscattered electron images of the fabricated TiB_2_–Ti composites are presented in Figure 6b–e. The SEM microstructure of the T2.5 sample (Figure 6b) shows the in-situ formed whiskers (pointed with red arrows) and TiB_2_ particles on the image (pointed with white arrows). In addition, a few unreacted TiB_2_ particles were observed in the microstructure, possibly due to the reaction between TiB_2_ and Ti. Previous investigations reported that the reaction between TiB_2_ and Ti during SPS processing might have been slowed down by the coarse TiB_2_ particles or the short holding time [43]. In the T5.0 sample, in-situ whiskers, particles, and unreacted particles were observed in the SEM microstructure. Furthermore, the T7.5 and T10 samples revealed the agglomerations of the in-situ particles because of the addition of the high-volume fraction of TiB_2_. The presence of whiskers was observed from the SEM micrographs, as indicated with red arrows, because of the reaction between Ti and TiB_2_. Furthermore, some unreacted TiB_2_ particles and TiB corona were noticed in the images due to the incomplete reaction between TiB_2_ and Ti, according to the investigation by Namini et al. [43] on the sintering of titanium composites.

### 3.5. Evaluation of the Microhardness and Tensile Properties

#### 3.5.1. Vickers Microhardness

The microhardness value of the sintered samples is shown in Figure 7. The T0.0 sample attained the hardness result of 188.1 HV1. In contrast, the T2.5, T5.0, T7.5, and T10 samples attained the hardness values of 228.5 HV1, 242.7 HV1, 262.4 HV1, and 304.8 HV1. The hardness values of the composites improved substantially with the reinforcement addition, attributable to the high hardness of the reinforcement [44]. Furthermore, reinforcement contents, homogeneity, and relative density are the factors that can affect the microhardness value of sintered materials [38]. Furthermore, the sintered composites’ hardness enhancement may be credited to the hindrances in dislocation motion caused by the in-situ-formed phases in the composites [45].

#### 3.5.2. Tensile Properties

The stress-against-strain curve in Figure 8a was used to determine the tensile strength (UTS) and the elongation values of the sintered samples. The stress of the sintered samples increased with the strain until plastic deformation occurs. The UTS value of the sintered titanium (T0.0 sample, Figure 8b) was found to be 433.8 MPa, while the UTS values of the sintered T2.5, T5.0, T7.5, and T10 composites were found to be 514.3 MPa, 443.8 MPa, 265.3 MPa, and 289.9 MPa, respectively. The UTS of the sintered T2.5 and T5.0 samples is higher than the UTS of the matrix sample (T0.0). This could be linked to good dispersion of the TiB_2_ particles, which blocked the dislocation movement in the composites, therefore enhancing the strengthening effect at the cost of its ductility [46]. However, in the TiB_2_-reinforced Ti composite, there are several ways in which the strengthening process can be achieved. It may be through a load transfer from the matrix to the reinforcement, or from an in-situ whisker formation resulting in solid solution strengthening, grain refinement, etc. Consequently, TiB_2_–Ti composites achieved a higher UTS than the matrix metal as a result of good interfacial bonding between the reinforcement and the matrix, high hardness of the reinforcement (TiB_2_), and the stress transfer from the matrix to the whiskers [43]. In contrast, an exception was noticed for T7.5 and T10 composites. According to Yan et al. [38], such trends in the tensile values may be caused by non-homogeneity in the microstructures as the reinforcement increases. Furthermore, the T0.0 sample had an elongation value of 27.1%. Meanwhile, T2.5, T5.0, T7.5, and T10 composites had the elongation values of 15.0%, 20.4%, 17.9%, and 10.6%, respectively. Achieving excellent ductility in composites depends on the mechanical properties and the evolution of the microstructures, which may be ascribed to good interfacial bonding, dispersion strengthening, grain refinement, and the coefficient of thermal expansion mismatch between the matrix and the reinforcement [46]. However, the elongation values of the composite decreased from the T5.0 sample from 20.4% to 10.6% with increasing TiB_2_ reinforcement. The reduction in tensile elongation may be ascribed to the clustering of ceramic particles in the composites’ microstructures. Therefore, as the volume fraction of TiB_2_ increases in the Ti matrix, this affects the resistance to plastic deformation of the fabricated composites, as noticed in the tensile elongation.

### 3.6. Nanomechanical Properties

#### 3.6.1. Load Displacement Curve

Figure 9a shows the load-displacement curve of all the sintered samples at 25 mN load. The curve displayed the loading and unloading cycle of each sintered sample. In addition, the curves appear smooth except for the T0.0 and T2.5 samples, which showed pop-in effects, suggesting a strain transfer and nucleation of dislocation at the grain boundaries [47]. A significant decrease in the penetration depth was observed from the curve due to the addition of the reinforcement. The T0.0 sample showed the highest penetration depth, followed by the sintered composites. This behavior suggests that sintered composites had more resistance to plastic deformation during the indentation process, therefore showing an enhancement in stiffness and hardness due to load transfer from the matrix to the ceramic reinforcement [48]. Meanwhile, Figure 9b shows the depth of the sintered samples at 25 mN as a function of time. The T0.0 sample was observed with a maximum penetration depth of 488 nm. Meanwhile, the sintered composites T2.5, T5.0, T7.5, and T10 had penetration depths of 479 nm, 455 nm, 414 nm, and 456 nm, respectively. It was found that the penetration depths of the composites showed a greater decrease than the T0.0 sample. The reduction in the penetration depths of the composites indicates their resistance to indentation load [49].

#### 3.6.2. Reduced Elastic Modulus (REM) and Hardness Values

The mean nano hardness and REM values of all the sintered samples were estimated from the nanoindentation data, as shown in Figure 10. The T0.0 sample had a mean nano hardness of 6034 MPa when the load was 25 mN, while the sintered composites (T2.5, T5.0, T7.5, and T10) were found with the following mean nano hardness values (6220 MPa, 7656 MPa, 9841 MPa, and 7349 MPa, respectively). The sintered composites had an improved nano hardness compared to the sintered titanium (T0.0) due to the addition of reinforcements. Furthermore, the nano hardness improved as the TiB_2_ reinforcement increased, except for the T10 sample, which experienced a sharp drop in nano hardness. This trend was also observed from the relative density of the T10 sample (Figure 2), which may be due to the high amount of the reinforcement. In addition, the mean-reduced elastic modulus of the sintered samples was determined from the nanoindentation data. The sintered titanium (T0.0 sample) was observed with a reduced elastic modulus value of 156 GPa. Meanwhile, T2.5, T5.0, T7.5, and T10 composites had reduced elastic modulus of 158 GPa, 159 GPa, 188 GPa, and 168 GPa, respectively. The elastic modulus values of the sintered composites showed a similar trend with the nano hardness results, suggesting that TiB_2_ imparts more stiffness in the sintered composites by improving their mechanical properties compared to the sintered Ti matrix.

### 3.7. Wear Behavior of the Sintered Samples

#### 3.7.1. Analysis of the Coefficient of Friction (COF)

The variety in COF against time for the sintered samples at the applied loads of 9 N and 13 N is shown in Figure 11a, b. As observed in Figure 11a, the COF of the T0.0 sample at the applied load of 9 N started from a momentary stage before achieving steady state friction between 0.40 and 0.54. In addition, the COF of the sintered composites started from a transient stage before attaining steady-state friction. The COF of the T2.5 sample fluctuates at the early stage around 450 s, possibly due to debris and tribo product formation [50], before becoming steady. However, T5.0, T7.5, and T10 composites experienced similar COF behaviors, and they became steady between 0.49 and 0.51. At the applied load of 13 N, the COF against time for the sintered samples is shown in Figure 11b. All the sintered samples exhibited similar trends in COF before achieving steady-state friction between 0.65 and 0.75. It was observed that the T2.5 sample had an early rise in COF amongst the other samples before it reached a steady state friction due to the contact between the counterpart surface and the sample. The fluctuations in the COF observed for all the sintered samples may be due to particle interplay and the localized fracture of the transfer layer at the sliding interface [51]. The COF of a material is related to the frictional force between the sliding body, signifying the influence of sliding velocity and load over a defined distance. This explains that the COF depends majorly on the load and sliding velocity. However, as observed in Figure 11a, b, the sintered samples under the applied load of 13 N experienced more wear than the sintered samples at 9 N due to the increment in the COF.

#### 3.7.2. Effect of the TiB_2_ on the Mean COF

The mean COF of the sintered samples was observed in Figure 12 at varied loads of 9N and 13 N. At the 9 N applied load, the T0.0 sample had the highest mean COF of 0.5352, which could be associated with its low-work hardening behavior and ductile nature [52]. Meanwhile, the sintered composites of T2.5, T5.0, T7.5, and T10 were observed with the COF of 0.5231, 0.5215, 0.5001, and 0.5171, respectively. The presence of the TiB_2_ particle was responsible for the lower COF observed in the sintered composites due to good adhesion between the Ti matrix and TiB_2_ particle [53]. It was also observed that the COF of the sintered composites declined with increasing volume fraction of TiB_2_, except for the T10 composite, which showed a slight increment in COF.

At 13 N applied load, the T0.0 sample had a COF of 0.7756 while the sintered composites T2.5, T5.0, T7.5, and T10 were observed with the COF of 0.7897, 0.7570, 0.7307, and 0.7446, respectively. It was noticed that the COF of the T2.5 composite was slightly higher than the COF of the T0.0 sample. This observation may be caused by the destruction of the oxide layer in the T2.5 composite due to the impact of the high load, thereby resulting in high COF. However, the sintered composites’ COF decreased with increasing TiB_2_ content, except for the T10 composite, which showed a slight increment in COF. This phenomenon was observed for the T10 composite under both loads, and this may be attributed to the microstructure evolutions, high hardness, and relative density of the composite [54]. Comparatively, the sintered samples’ COF increased with increasing load from 9 N to 13 N. It is possible that at 13 N load, there was a rise in contact temperature, which produced additional heat due to area contact between the specimen and counterface [55].

#### 3.7.3. Effect of the TiB_2_ on the Wear Rate and Depth

The wear rate of the sintered samples subjected to the applied loads of 9 N and 13 N is shown in Figure 13. As observed, the wear rate for all the sintered samples increases as the applied load increases. This shows that wear depth is directly related to the applied load during the wear experiment for the sintered samples. At 9 N and 13 N, the T0.0 sample had the highest wear rate, while the sintered composites’ wear rate decreased with increasing TiB_2_ contents. This indicates that the sintered composites experienced better wear resistance than the T0.0 sample. Alaneme et al. [55] presented a similar investigation wherein the enhanced wear resistance was attributed to the high hardness of the reinforcements. The wear enhancement in the sintered composites can also be linked to the good load-bearing capacity of TiB_2_ by restricting the fracture and plastic deformation of the sintered composite [56,57].

Figure 14a–d shows the profilometer’s wear track depth of T0.0 and T10 samples at varied loads. Figure 14a,b shows the wear track depths of the T0.0 sample under 9 N and 13 N. It can be seen that wear is more predominant when the samples were subjected to the applied load of 13 N than 9 N. Figure 14c,d shows the wear track depths of the T10 composite under both loads, indicating similar trends to the T0.0 sample. It can be concluded that the wear depth is directly proportional to the applied load [58].

#### 3.7.4. Morphology of the Worn Surface and Debris

Figure 15a–e shows the morphology of the worn surface of sintered samples subjected under 9 N, and at the linear velocity of 5.17 cm/s. The worn surface of the T0.0 samples shows the evidence of grooves along the gliding direction due to the contact areas between the ball (counterface) and the sample. As observed, delamination, which is in the form of a sheet-like detachment, is visible on the image due to ploughing, and some of the worn debris adhered to the surface of the sample. The worn surface features observed for the T0.0 sample can be ascribed to their low hardness and ductile nature [52]. Therefore, the T0.0 sample is perceived to have shown both abrasive and adhesive wear mechanisms. Meanwhile, the worn surfaces of the composites (T2.5, T5.0, T7.5, and T10) are shown in Figure 15b–e. Grooves can be noticed along the gliding directions of the composites, and compaction of wear debris and delamination was observed from the composites’ morphologies. These worn surface features are similar to that of the T0.0 sample, indicating both abrasive and adhesive wear mechanisms. The presence of the hard TiB_2_ particles in the composites helped in enhancing better wear resistance. As the load increased to 13 N (Figure 16a–e), deep grooves were seen in the T0.0 sample (Figure 16a) due to the penetration of the counterface on the specimen due to plastic deformation, which resulted in severe delamination [53]. However, the composites were also micro-ploughed at higher loads, resulting in delamination and fractured particles. The wear debris adhered to the surface of the composites during the wear test, and the presence of the hard TiB_2_ particles contributed to attaining shallow grooves in the composites. All the observed wear mechanisms indicate both abrasive and adhesive wear.

#### 3.7.5. Morphology of the Wear Debris

The morphology of the debris collected during the tribology test of the fabricated samples subjected under 9 N is shown in Figure 17a–e. The SEM morphology of the wear debris of the T0.0 sample (Figure 17a) shows large and small flakes alongside fragmented particle debris. This suggests that delamination wear occurred during the tribology test [59,60]. Meanwhile, the fabricated composites T2.5 in Figure 17b had a mixture of large and small flakes with fragmented debris. The fabricated composites T5.0, T7.5, and T10 in Figure 17c–e exhibited similar phenomena to the fabricated T2.5 sample. Though some of the flattened flakes were decreased, it was observed that as the reinforcement content increased, the fragmented particles became smaller. This indicated that the delaminating wear mechanism was moderate during the tribology test due to the hardness enhancement by the reinforcement [60,61]. On the other hand, the morphology of the wear debris subjected under 13 N is shown in Figure 18a–e. As observed in the micrographs (Figure 18a), the wear debris showed a similar wear mechanism to the previous fabricated samples (Figure 17a). Furthermore, the wear debris in Figure 18b–e had a mixture of large and small flakes. This indicates some similarity with the wear mechanism of the fabricated composites in Figure 17b–e. However, it was observed that the fabricated samples seem to have more agglomerated particles, possibly due to the impact of the counterface sliding on the specimen at a very high load.

### 3.8. Fractography

Figure 19 shows the fractured surfaces of the sintered samples. The fractured T0.0 sample depicts a ductile fracture behavior because of the dimples (depicted with red arrows) observed in the micrograph (Figure 19a), which was buttressed by the tensile behavior of the sample in Figure 19a. The fractured T2.5, T5.0, T7.5, and T10 composites in Figure 19b–e were observed with a combination of cleavages, tiny-needle whiskers, and large cracks (depicted with white arrows), which symbolized a brittle behavior and dimples (ductile behavior). The fractured surfaces of the sintered composites do not reveal pull-outs of TiB2 reinforcement, indicating a clean interface between the reinforcement and the matrix metal [43].

## 4. Conclusions

The titanium composites were consolidated via the SPS process at 1050 °C. The microhardness, relative densities, nanomechanical properties, tensile properties, microstructure evolution, and fractured surfaces were examined. In this study, adding TiB_2_ reinforcement to the titanium matrix improved the sintered specimens’ relative densities from 97.5% to 99.3%. The Vickers microhardness values of the sintered sample increased from 188.1 HV1 to 304.8 HV1, about a 62% increment. The highest UTS value (514.3 MPa) was observed for the fabricated T2.5 composites before the UTS decreased with the volume of TiB_2_ reinforcement. The highest elongation value (27.1%) was observed for the sintered T0.0 matrix, and a decrease in elongation values was noticed for the sintered composites. This proved that the tensile strength of the sintered composites increased at the expense of their elongations. The nanohardness improved from 6034 MPa to 9841 MPa, while the reduced elastic modulus improved from 156 GPa to 188 GPa upon adding TiB_2_ content. However, a slight drop was noticed for the T10 sample, possibly due to agglomeration. The result of the X-ray diffractograms showed the evolution of new phases, and there was no formation of the TiB_2_ phase in the sintered composites’ XRD spectrum due to the complete reaction between the Ti and TiB_2_. The microstructures displayed lamellar structures with the distribution of whiskers, unreacted particles, and TiB corona. The sintered composites had better wear resistance than the T0.0 sample. The fractured surfaces of the sintered composites showed a combination of brittle and ductile behavior, while the sintered titanium showed ductile behavior.

## Figures and Tables

**Figure 1 materials-16-02078-f001:**
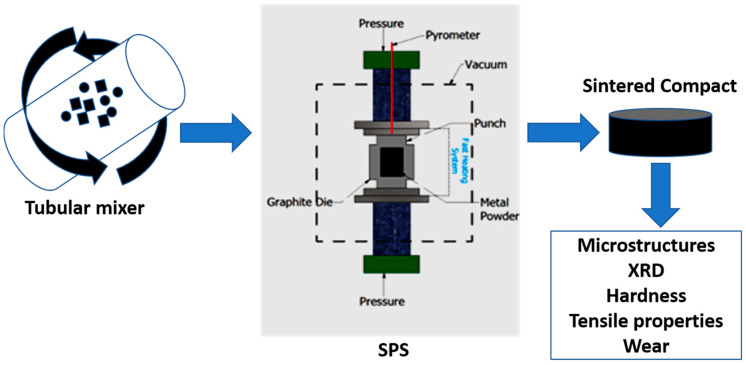
Graphical illustration of the experiment.

**Figure 2 materials-16-02078-f002:**
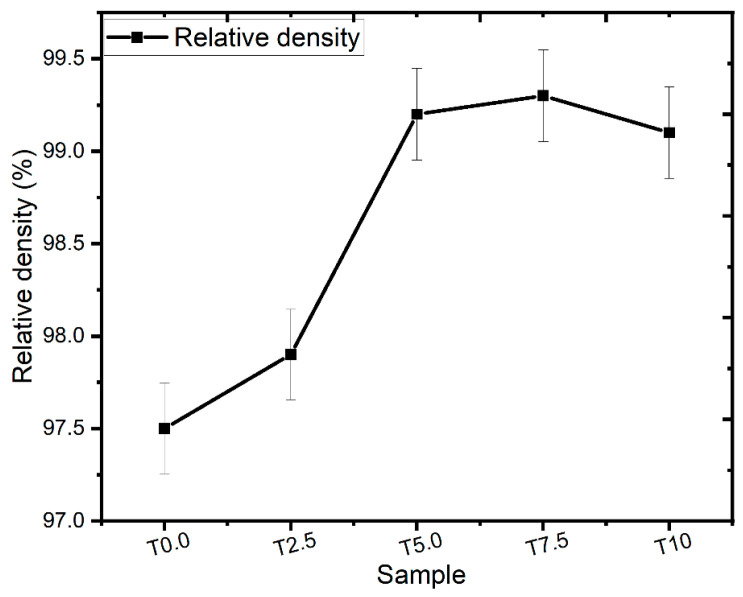
The relative densities of the consolidated Ti and TiB_2_–Ti composites.

**Figure 3 materials-16-02078-f003:**
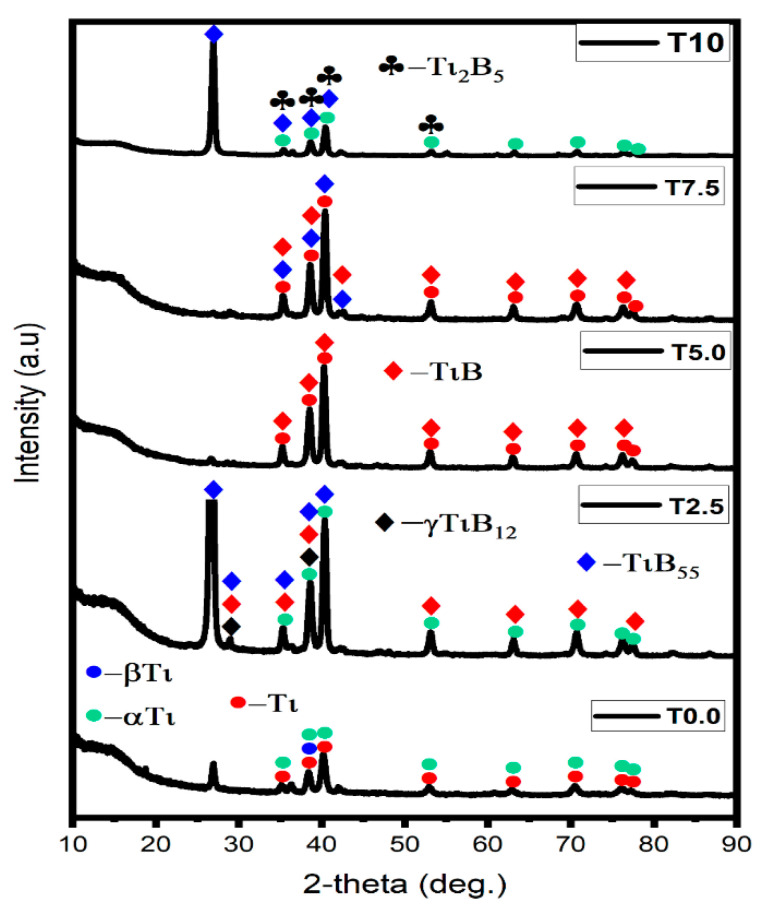
The diffractogram of the consolidated Ti and TiB_2_–Ti composites.

**Figure 4 materials-16-02078-f004:**
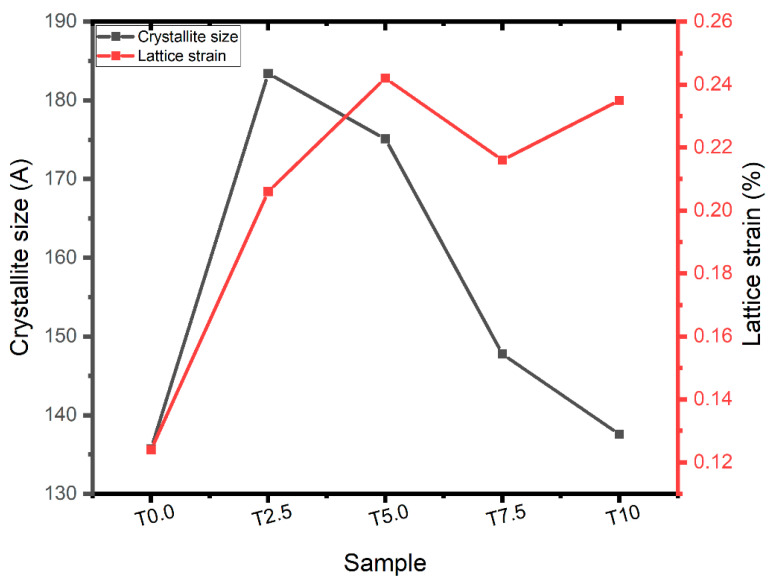
Values of the crystallite size and lattice strain of the consolidated Ti and Ti-TiB_2_ composites.

**Figure 5 materials-16-02078-f005:**
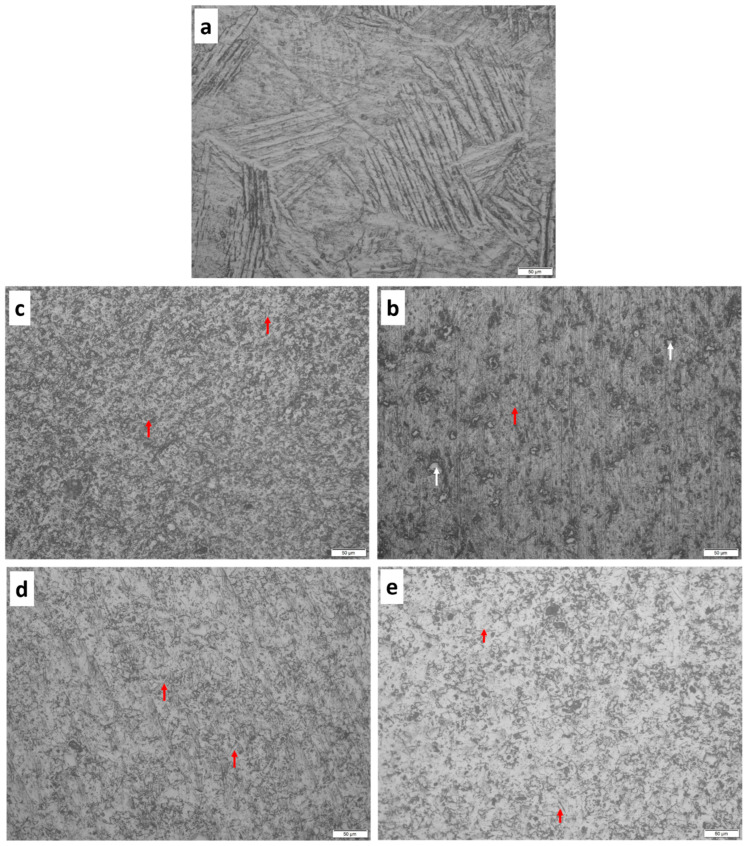
Optical images of the sintered samples (**a**) T0.0, (**b**) T2.5, (**c**) T5.0, (**d**) T7.5, and (**e**) T10.

**Figure 6 materials-16-02078-f006:**
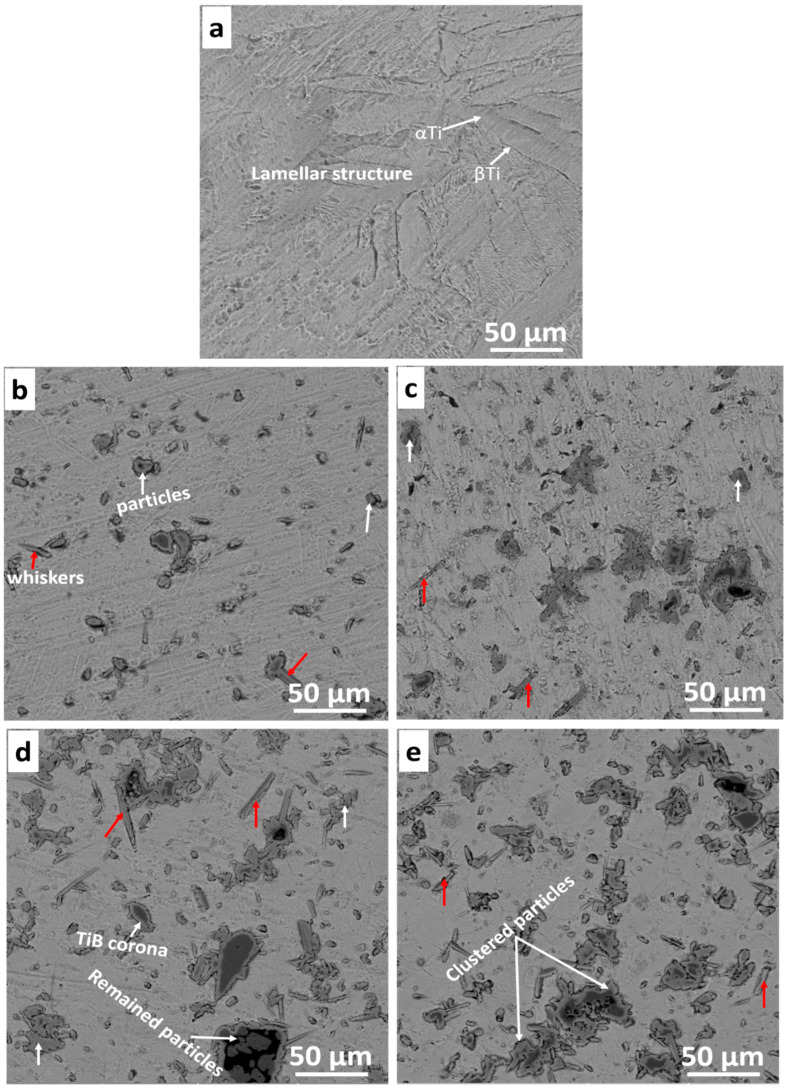
Secondary electron images of sintered samples (**a**) T0.0, (**b**) T2.5, (**c**) T5.0, (**d**) T7.5 (**e**) T10.

**Figure 7 materials-16-02078-f007:**
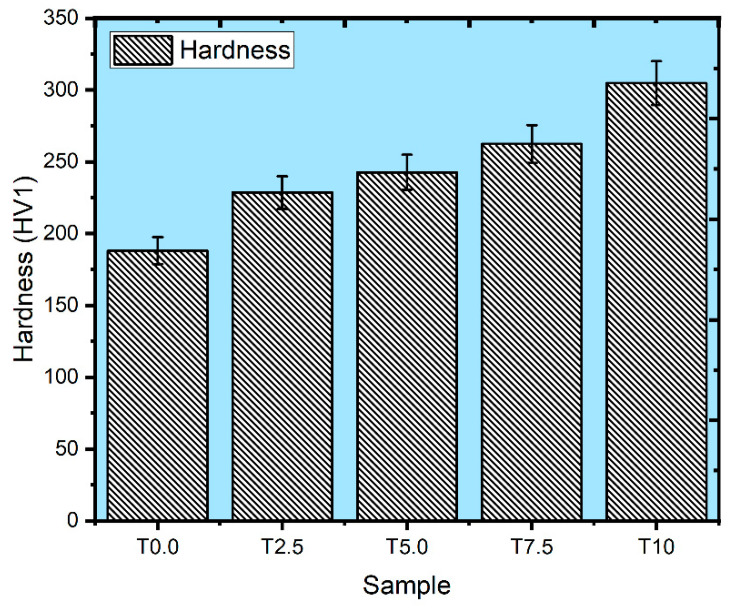
Vickers hardness of sintered samples.

**Figure 8 materials-16-02078-f008:**
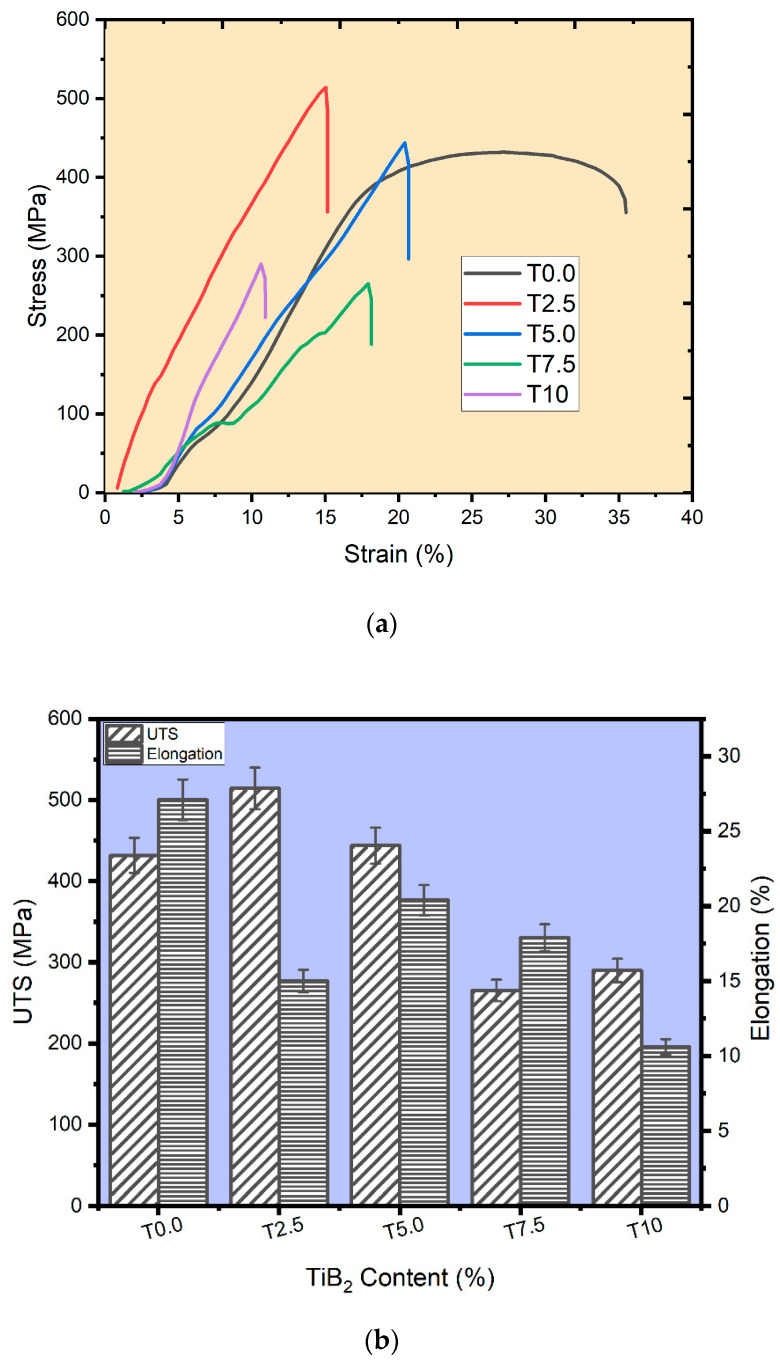
(**a**) The stress-strain curve of the sintered samples. (**b**) The UTS and Elongation plot of the sintered samples.

**Figure 9 materials-16-02078-f009:**
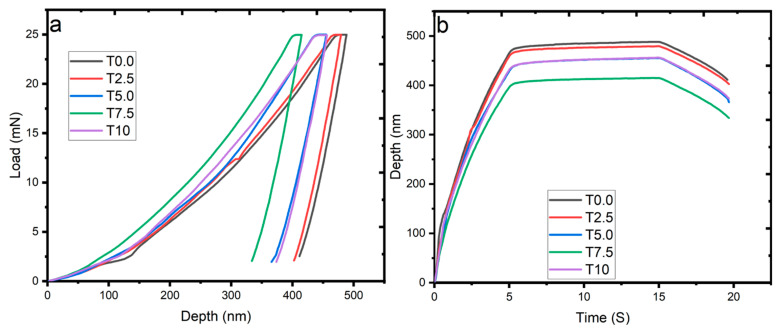
The nanoindentation curves of the sintered samples at the load of 25 mN: (**a**) Load-displacement, and (**b**) Penetration depth as a function of time curves.

**Figure 10 materials-16-02078-f010:**
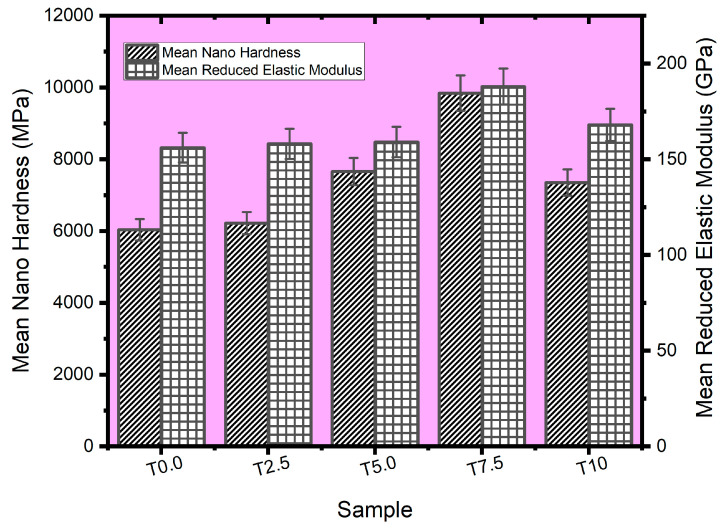
The mean hardness and reduced elastic modulus plots of the sintered samples at the indentation load of 25 mN.

**Figure 11 materials-16-02078-f011:**
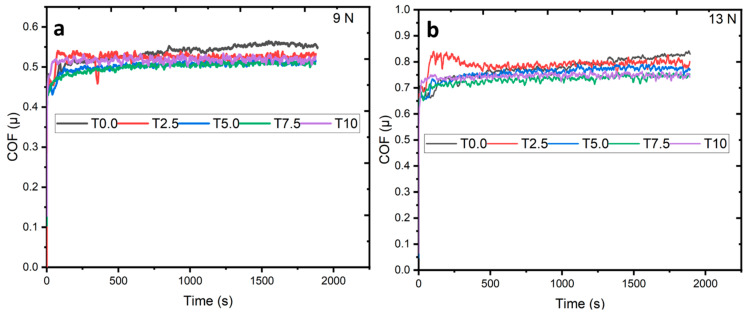
The variation of COF against time of the sintered samples under the applied load (**a**) 9 N and (**b**) 13 N.

**Figure 12 materials-16-02078-f012:**
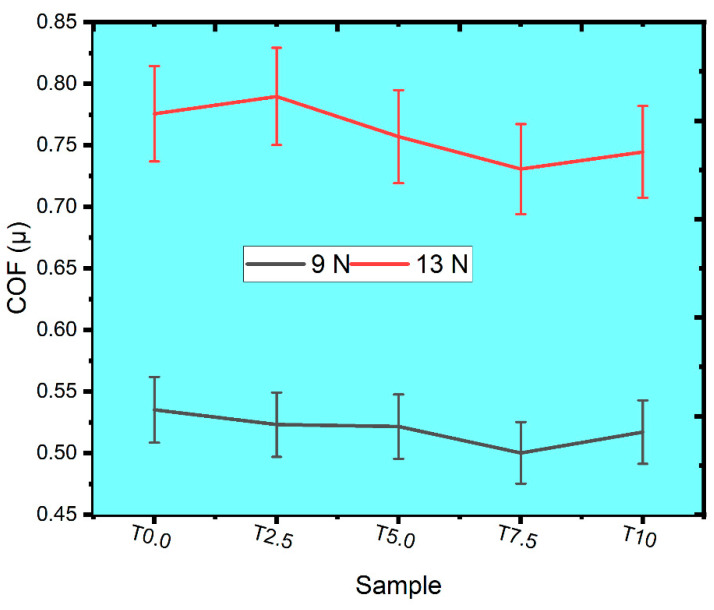
The plot of the mean COF for the sintered samples under 9 N and 13 N loads.

**Figure 13 materials-16-02078-f013:**
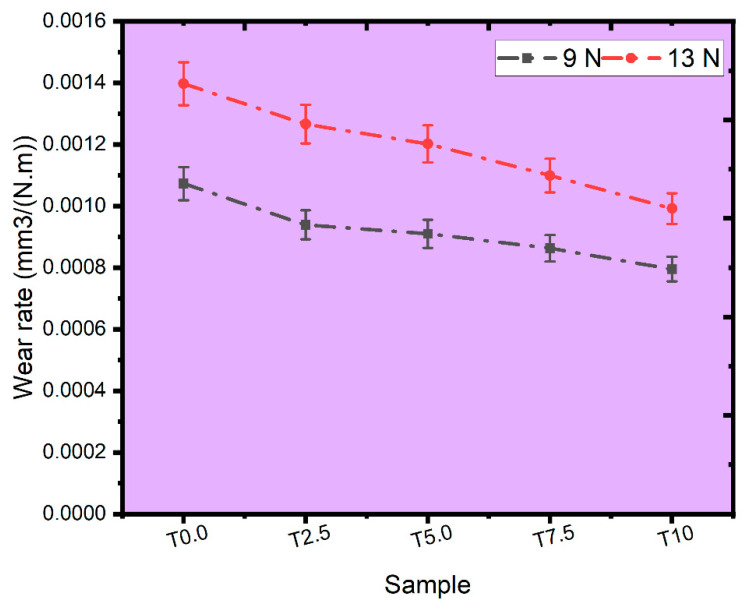
The plot of the wear rate for the sintered samples under 9 N and 13 N loads.

**Figure 14 materials-16-02078-f014:**
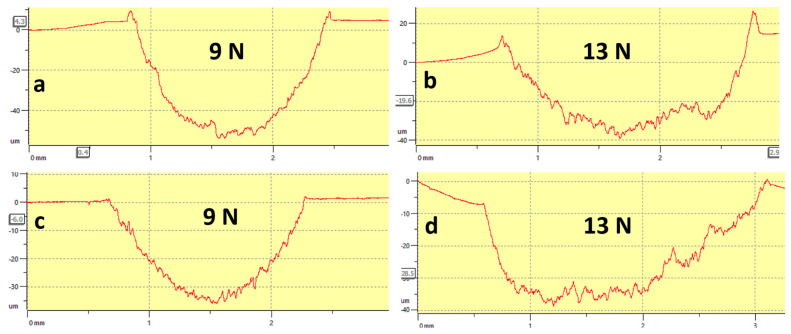
The profilometer’s wear depth of the sintered samples (**a**,**b**) T0.0 and (**c**,**d**) T10.

**Figure 15 materials-16-02078-f015:**
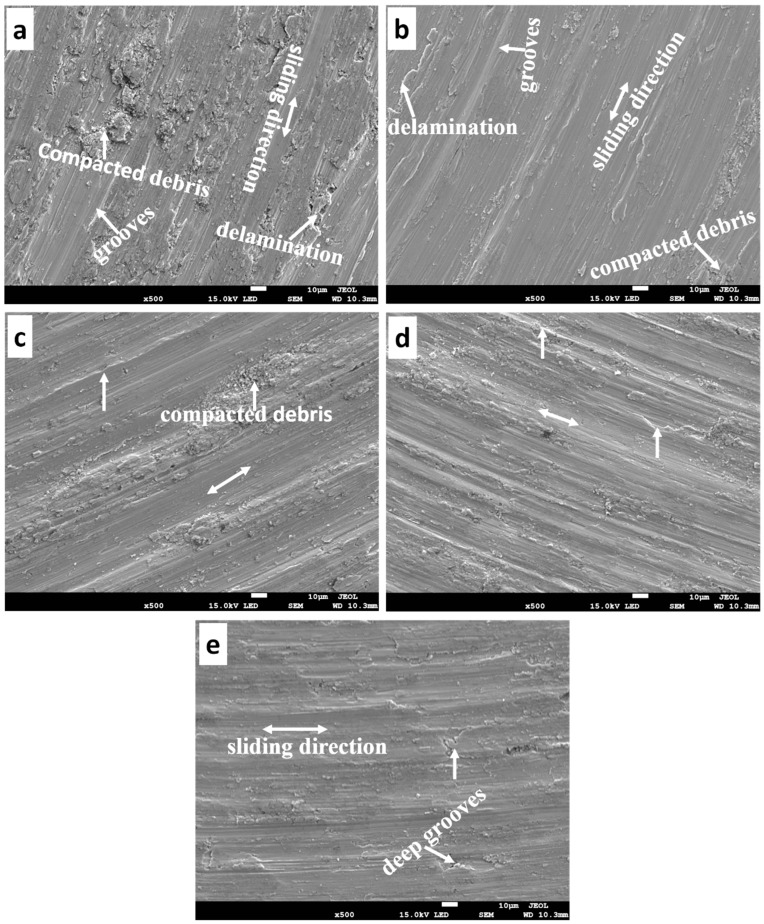
The worn surface of the sintered samples under 9 N (**a**) T0.0, (**b**) T2.5, (**c**) T5.0, (**d**) T7.5, and (**e**) T10.

**Figure 16 materials-16-02078-f016:**
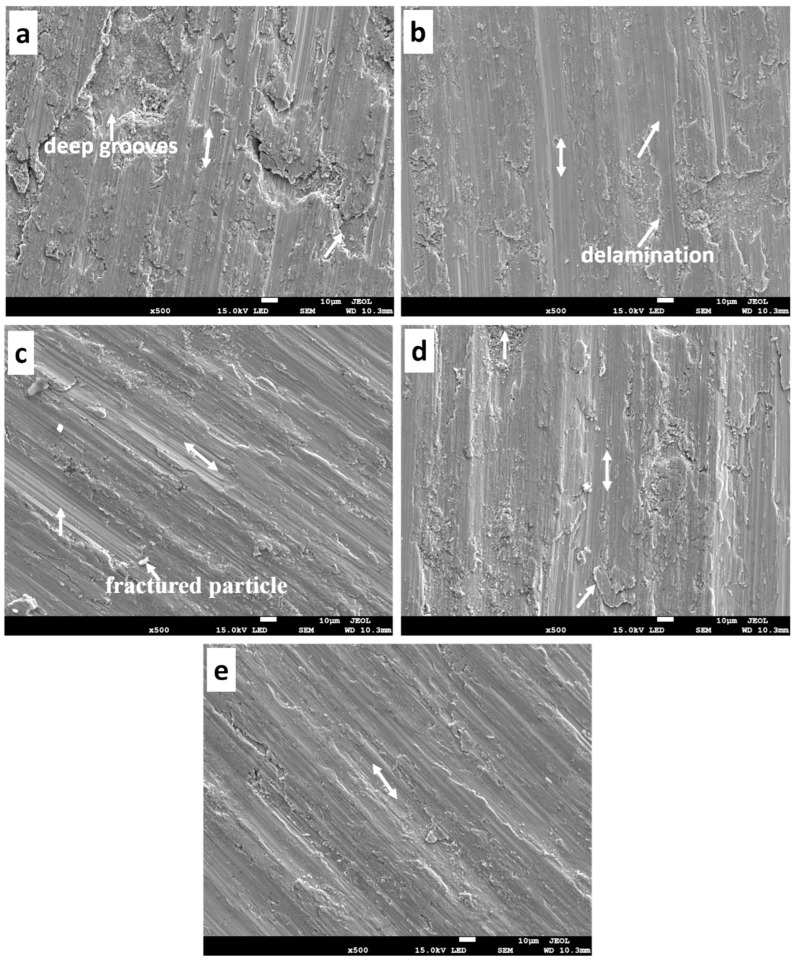
The worn surface of the sintered samples under 13 N (**a**) T0.0, (**b**) T2.5, (**c**) T5.0, (**d**) T7.5, and (**e**) T10.

**Figure 17 materials-16-02078-f017:**
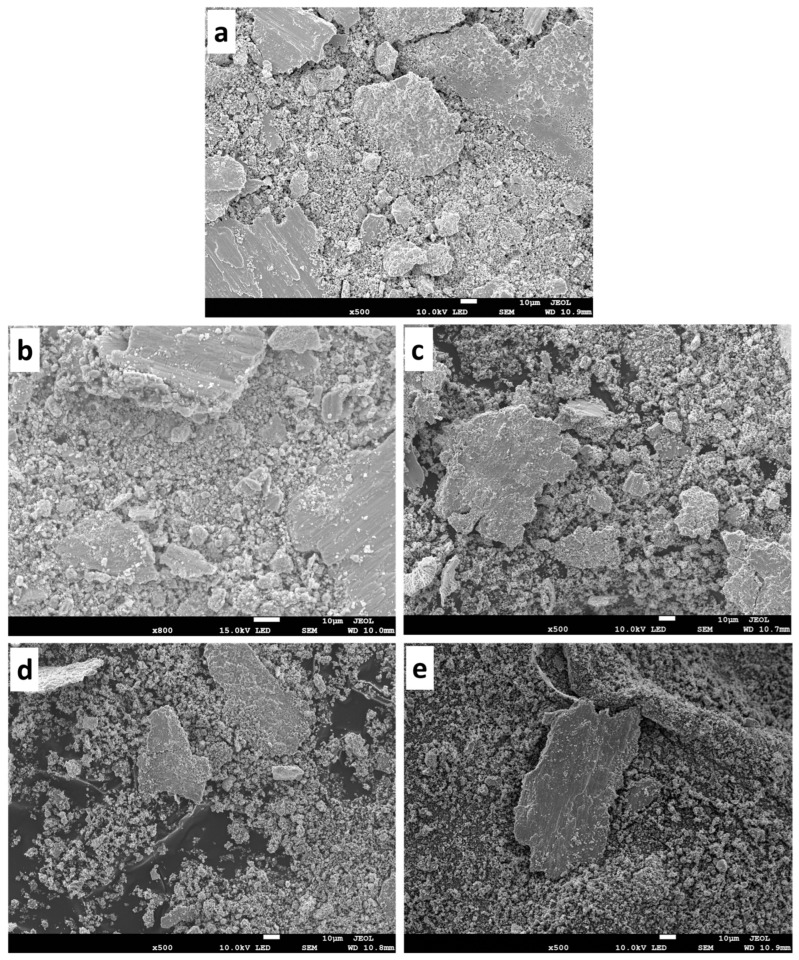
The wear debris of the sintered samples under 9 N (**a**) T0.0, (**b**) T2.5, (**c**) T5.0, (**d**) T7.5, and (**e**) T10.

**Figure 18 materials-16-02078-f018:**
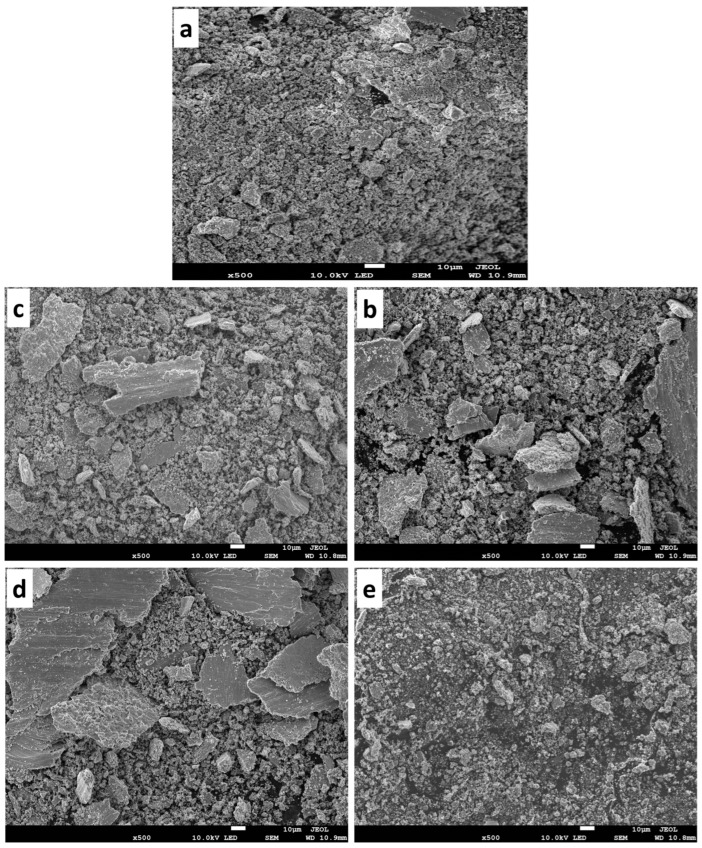
The wear debris of the sintered samples under 13 N (**a**) T0.0, (**b**) T2.5, (**c**) T5.0, (**d**) T7.5, and (**e**) T10.

**Figure 19 materials-16-02078-f019:**
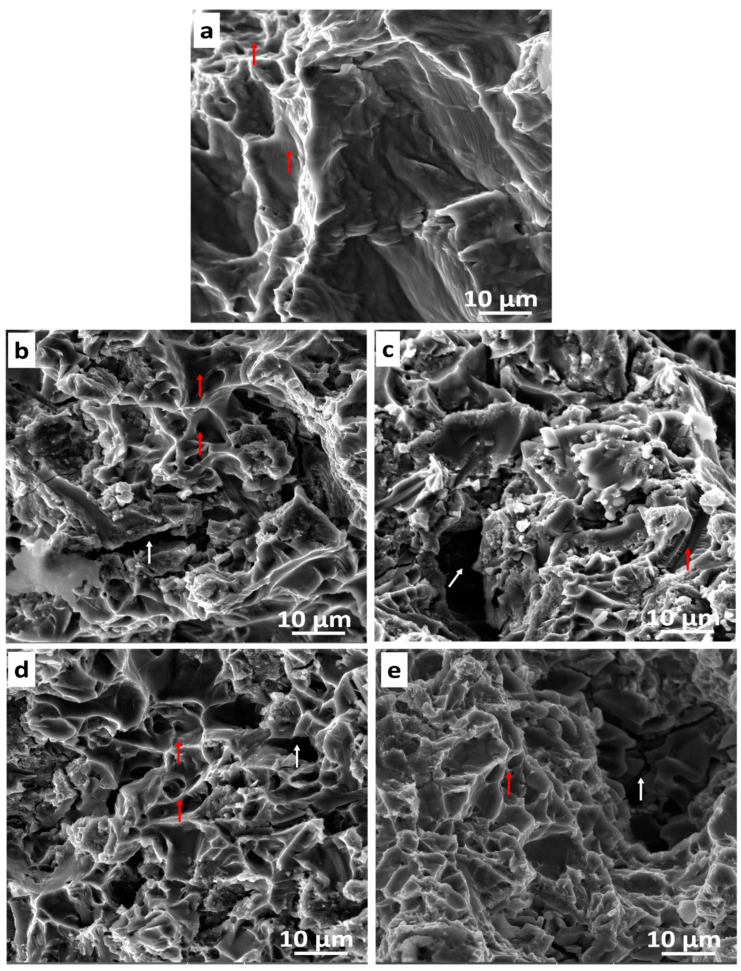
The micrographs of the fractured surfaces of the sintered samples (**a**) T0.0, (**b**) T2.5, (**c**) T5.0, (**d**) T7.5, and (**e**) T10.

**Table 1 materials-16-02078-t001:** Sintered sample codes.

Samples	Ti	Ti-2.5 wt.% TiB_2_	Ti-5.0 wt.% TiB_2_	Ti-7.5 wt.% TiB_2_	Ti-10 wt.% TiB_2_
Codes	T0.0	T2.5	T5.0	T7.5	T10

## Data Availability

The data utilized for this study are available upon request.

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
