# Peer review of "Evaluation of the Wear and Mechanical Properties of Titanium Diboride-Reinforced Titanium Matrix Composites Prepared by Spark Plasma Sintering"

_materials, 2023, doi:10.3390/ma16052078_

Round 1

Reviewer 1 Report

This article needs to be improved in the following aspects:1. English needs to be further improved, 2. The pictures are very rough, and need to be improved to the quality of each picture, 3. Highlight the scientific innovation points. 4. For the lack of discussion of the current research status, the author needs to quote more of the previous work to reflect the innovation of this paper. The article should be reconsidered after careful modification.

https://doi.org/10.1016/j.apsusc.2022.152495

https://doi.org/10.1016/j.matdes.2016.03.061

Author Response

Reviewers’ Comments

Thank you so much for the privilege of reviewing our manuscript titled " Evaluation of the spark plasma sintering of titanium diboride-reinforced titanium matrix composites’’. The answers to the Reviewers' comments are in red font. Corrections have been made in the manuscripts where necessary.

#1

This article needs to be improved in the following aspects:

  1. English needs to be further improved,

Answers: The English has been improved using Grammarly.

  1. The pictures are very rough, and need to be improved to the quality of each picture,

Answers: Most of the images were originally obtained from the scanning electron microscope and they have been improved.

  1. Highlight the scientific innovation points.

Answers:

  • The excellent adhesion between the Ti and TiB2 particle lower the friction coefficient of the sintered composites.
  • The tensile properties of the sintered composites improved at the expense of their elongations.
  • The wear rate of all the sintered samples increased with increasing applied load.
  • The mechanical properties of the sintered composites exhibited a sharp drop with a high-volume fraction of TiB2 particles.
  1. For the lack of discussion of the current research status, the author needs to quote more of the previous work to reflect the innovation of this paper. The article should be reconsidered after careful modification.

https://doi.org/10.1016/j.apsusc.2022.152495,

https://doi.org/10.1016/j.matdes.2016.03.061    

Answers: The reference above have been considered.

Reviewer 2 Report

I can recommend the publication of this manuscript after a minor revision.

1. Write keywords in alphabetical order.

2. Line 42: minor mistake. “ Singh et al. [14] also showed....” Ref. 14 is: “C. Cai, B. Song, P. Xue, Q. Wei, J. Wu, W. Li, Y. Shi,...”

3. Line 58: minor mistake: “[18].The process...”

4. Line 62: “..., etc. [21–25].”

You will likely need to re-write your citation sentences, rather than simply replacing the numbers with Authors’ names. This is due to the fact that in order to give readers the maximum appreciation of how your work builds on previous results, each one of the cited sources should be discussed individually and explicitly to demonstrate their significance to your study. We ask that you use the authors' surnames as the subject of a verb, and then state in one or two sentences what they claim, what evidence they provide to support their claim, and how you evaluate their work. We also, therefore, ask that you avoid citing more than one reference in one sentence. This will give you a chance to discuss each reference separately.

What we are asking for is something like this: “Smith (2011) describes the development of a finite element model of hot forging and claims excellent agreement between the model and experiments.  However, he tests only one operating condition, tunes his model by modifying the friction coefficient, and compares only the total tool force. A much more detailed comparison would be required to evaluate the precise conditions under which finite element modeling is truly accurate."

5. Line 77: insert the correct symbol for degrees.

6. Write all equations using a mathematical equation editor.

7. Line 139, Insert all the SEM parameters, such as magnification, acceleration voltage, working distance, and image pixel resolution (fig. 1).

8. Give more details about the statistical analysis applied in this manuscript (method, software, validation, and so on).

9. Specify the limits of this study. State in more detail the respective advantages and disadvantages.

10. Explain with more details sentences from lines: 214-216, 262-265, 307-309, 329-330, 400-403, 420-422.

11. Lines 464-465: ref. 1 incomplete. Must be added the following information: Publication: Metallurgical and Materials Transactions A, Volume 26, Issue 12, pp. 3211-3223. Pub Date: December 1995 DOI: 10.1007/BF02669450.

12. Line 510: minor mistake: “(English Lett. 31 (2018)...”

13. Line 564: minor mistake: “...and B 4 C Particles ...”

Even though the work is relevant to the journal's scope, i.e., Materials, I do not find even a single article published in the journal in the list of references.

If possible, I recommend the following references:

[1] https://doi.org/10.1016/j.matpr.2019.12.128

[2] Ş. Ţălu, Micro and nanoscale characterization of three-dimensional surfaces. Basics and applications. Napoca Star Publishing House, Cluj-Napoca, Romania, 2015.

This manuscript can be published after the mentioned revisions.

Author Response

#2

  1. Write keywords in alphabetical order.

Answer: Ceramic reinforcement, Microstructures, Nanomechanical properties, Tensile behavior, Titanium alloy, and Tribology

  1. Line 42: minor mistake. “Singh et al. [14] also showed....” Ref. 14 is: “C. Cai, B. Song, P. Xue, Q. Wei, J. Wu, W. Li, Y. Shi,...”

Answers: The reference has been fixed in the manuscript.

  1. Line 58: minor mistake: “[18]. The process...”

Answers: The reference has been fixed in the manuscript.

  1. Line 62: “..., etc. [21–25].”

You will likely need to re-write your citation sentences, rather than simply replacing the numbers with Authors’ names. This is due to the fact that in order to give readers the maximum appreciation of how your work builds on previous results, each one of the cited sources should be discussed individually and explicitly to demonstrate their significance to your study. We ask that you use the authors' surnames as the subject of a verb, and then state in one or two sentences what they claim, what evidence they provide to support their claim, and how you evaluate their work. We also, therefore, ask that you avoid citing more than one reference in one sentence. This will give you a chance to discuss each reference separately.

What we are asking for is something like this: “Smith (2011) describes the development of a finite element model of hot forging and claims excellent agreement between the model and experiments.  However, he tests only one operating condition, tunes his model by modifying the friction coefficient, and compares only the total tool force. A much more detailed comparison would be required to evaluate the precise conditions under which finite element modeling is truly accurate."

Answers: Thank you for your comment. Some of the citations have been described according to the format above. However, the referencing style used by the authors differs from the suggested one. For example: ‘’Alaneme et al. [54] presented a similar investigation wherein the enhanced wear resistance was attributed’’………

  1. Line 77: insert the correct symbol for degrees.

Answers: This has been fixed in the manuscript.

  1. Write all equations using a mathematical equation editor.

Answers: Mathematical editor has been used in the manuscript.

  1. Line 139, Insert all the SEM parameters, such as magnification, acceleration voltage, working distance, and image pixel resolution (fig. 1).

Answers: This has been fixed in the manuscript.

  1. Give more details about the statistical analysis applied in this manuscript (method, software, validation, and so on).

Answers: The statistical analysis commenced with data collection and analyzing them after the experimentation process. The experimental procedure has been explained in the manuscript. During the analysis, most of the data were placed in a Microsoft excel and software like Origin Pro was used for most of the results interpretation. Results validation was later carried out and compared with the existing literatures.

  1. Specify the limits of this study. State in more detail the respective advantages and disadvantages.

Answers: The study limits include the inability to use transmission electron microscopy (TEM) to analyze the specimens, and the process of mixing the raw materials could have been changed to ball milling to assess the homogeneity of the materials. The reaction between the Ti and TiB2 could have been evaluated with HSC Chemistry software to understand the enthalpy and Gibbs free energy of the material.

  1. Explain with more details sentences from lines: 214-216, 262-265, 307-309, 329-330, 400-403, 420-422.

Answers: This has been revised in the manuscript.

  1. Lines 464-465: ref. 1 incomplete. Must be added the following information: Publication: Metallurgical and Materials Transactions A, Volume 26, Issue 12, pp. 3211-3223. Pub Date: December 1995 DOI: 10.1007/BF02669450.

Answers: Thank you, it has been revised in the manuscript.

  1. Line 510: minor mistake: “(English Lett. 31 (2018).”

Answers: it has been corrected in the manuscript.

  1. Line 564: minor mistake: “...and B 4 C Particles ...”

Answers: it has been corrected in the manuscript.

Even though the work is relevant to the journal's scope, i.e., Materials, I do not find even a single article published in the journal in the list of references.

If possible, I recommend the following references:

[1] https://doi.org/10.1016/j.matpr.2019.12.128 (This reference has been considered in the manuscript.)

[2] Ş. Ţălu, Micro and nanoscale characterization of three-dimensional surfaces. Basics and applications. Napoca Star Publishing House, Cluj-Napoca, Romania, 2015.

This manuscript can be published after the mentioned revisions.

Reviewer 3 Report

This manuscript demonstrates the preparation of a Ti-based composite material formed by spark plasma sintering (SPS) technique staring from Ti and TiB2 as raw material. The material seem to show high mechanical properties; however, because the authors have carried out SPS in the same material system (Ti-TiB2) in a lower temperature: 900℃,  (https://doi.org/10.1016/j.jmrt.2022.02.048), the present manuscript is not evaluating the effect of SPS as stated in the title but is evaluating the structure and performance of the material sintered by SPS at a higher temperature: 1050℃. Considering this point. the experimental background and discussions are insufficient and the difference against their previous report must be fully explained. In addition, many results require additional information as mentioned afterwards. Finally, the reference number in the manuscript does not match the ones in the list at all. For example, the authors previous study introduced in line 39 says refers to [13] but it is probably [34] and the one in line 42 explaining Singh's work is not [14] but is probably [27]. Authors must revise carefully if to be resubmitted. From all these aspects, the manuscript cannot be accepted in the present form. Detail comments are as follows.

1. As mentioned beforehand, the authors have already considered SPS of Ti-TiB2 system at a lower temperature: 900℃, with 4 samples having same Ti/TiB2 content (0~7.5wt%) and have considered effect of TiB2 addition on density, tensile strength and Vickers microhardness. Detail of this study must be introduced in the introduction along with more previous literatures on this material system (e.g. https://doi.org/10.1016/j.msea.2007.01.161; https://doi.org/10.1016/j.matchemphys.2019.122556) and the novelty of the both the study and findings must be clearly explained in the introduction and results and discussions, respectively. In addition, if the authors are focusing on SPS, it should be compared with samples fabricated by normal sintering.

2. The authors are using the same raw materials as reported in their previous paper (https://doi.org/10.1016/j.jmrt.2022.02.048) and considering that the materials are purity above 99%, Figure 1 is not needed since they are introduced in your previous work. Especially, the SEM image Fig.1(a) is completely the same position as that in your previous study. Thus, usage should be avoided. If the authors want to evaluate the raw material, I believe it is better to evaluate after the mixing to ensure the homogeneity of the powder compact before the SPS.

3. The authors said that the relative density decreases for T10. However, it seems that error bar of T5.0, T7.5 and T10 are all in the similar range. Is this difference significant? Statical analysis should be conducted to ensure such small difference.

4. In the XRD shown in Figure 4, the result shows that Ti2B5 is formed at high TiB2 content. However, in your previous study, Ti2B5 is only detected at low TiB2 content of 2.5wt% and is not detected above 5.0wt%. The result seems to show an opposite trend. Please provide explanation on this matter.

5. To utilize the Williamson-Hall method to calculate the lattice strain, I believe we needs to confirm the linearity between sinθ and (Δ2θ)cosθ. How was the linearity? This information should be supplied.

6. In line 213, the authors have written that aggregates of TiB2 particles are observed. However, in the XRD patterns, no TiB2 is observed. Please provide explanation on this matter.

7. In the friction experiment, considering that TiB2 is more harder than the Ti matrix, I believe that usually the hard particles remain at the surface of the sliding track and reduces the contact area against the ball. However, I could not observe any TiB2 particles in the sliding track of the ball in contrast to Fig.6. Please provide explanation on this matter.

Author Response

#3

This manuscript demonstrates the preparation of a Ti-based composite material formed by spark plasma sintering (SPS) technique staring from Ti and TiB2 as raw material. The material seems to show high mechanical properties; however, because the authors have carried out SPS in the same material system (Ti-TiB2) in a lower temperature: 900℃, (https://doi.org/10.1016/j.jmrt.2022.02.048), the present manuscript is not evaluating the effect of SPS as stated in the title but is evaluating the structure and performance of the material sintered by SPS at a higher temperature: 1050℃. Considering this point. the experimental background and discussions are insufficient and the difference against their previous report must be fully explained. In addition, many results require additional information as mentioned afterwards. Finally, the reference number in the manuscript does not match the ones in the list at all. For example, the authors previous study introduced in line 39 says refers to [13] but it is probably [34] and the one in line 42 explaining Singh's work is not [14] but is probably [27]. Authors must revise carefully if to be resubmitted. From all these aspects, the manuscript cannot be accepted in the present form. Detail comments are as follows.

  1. As mentioned beforehand, the authors have already considered SPS of Ti-TiB2system at a lower temperature: 900℃, with 4 samples having same Ti/TiB2content (0~7.5wt%) and have considered effect of TiB2 addition on density, tensile strength and Vickers microhardness. Detail of this study must be introduced in the introduction along with more previous literatures on this material system (e.g. https://doi.org/10.1016/j.msea.2007.01.161; https://doi.org/10.1016/j.matchemphys.2019.122556) and the novelty of the both the study and findings must be clearly explained in the introduction and results and discussions, respectively. In addition, if the authors are focusing on SPS, it should be compared with samples fabricated by normal sintering.

Answers: Thank you for the comments. I appreciate it. Firstly, the authors agreed on the title of this manuscript because it was different from our previous manuscript published by JMRT ((https://doi.org/10.1016/j.jmrt.2022.02.048). The manuscript published by JMRT utilized and compared two various ceramics, and a few mechanical analyses were performed. The tensile results were evaluated using the hardness values. This present study was specifically carried out to assess titanium composites' mechanical and wear properties. All the citations and references have been corrected; the error occurred from the citing software. The suggested manuscripts have been considered. 

  1. The authors are using the same raw materials as reported in their previous paper (https://doi.org/10.1016/j.jmrt.2022.02.048) and considering that the materials are purity above 99%, Figure 1 is not needed since they are introduced in your previous work. Especially, the SEM image Fig.1(a) is completely the same position as that in your previous study. Thus, usage should be avoided. If the authors want to evaluate the raw material, I believe it is better to evaluate after the mixing to ensure the homogeneity of the powder compact before the SPS.

Answers: The authors repeated the images because the past manuscript showed the micrographs of two ceramics. We noticed that recently published papers had only used micrographs of raw powders and not milled powders, except if the manuscript focuses on the effect of milling.

  1. The authors said that the relative density decreases for T10. However, it seems that error bar of T5.0, T7.5 and T10 are all in the similar range. Is this difference significant? Statically analysis should be conducted to ensure such small difference.

Answers: The difference in the relative density from T5.0 composite to T10.0 is very minute (about 0.1 range difference). The difference is not too significant, which is the reason for the observed error bar. Error bar analysis on Origin software was explored for this calculation.

  1. In the XRD shown in Figure 4, the result shows that Ti2B5is formed at high TiB2content. However, in your previous study, Ti2B5 is only detected at low TiB2 content of 2.5wt% and is not detected above 5.0wt%. The result seems to show an opposite trend. Please provide explanation on this matter.

Answers: This could be as a result of the phase transformation that occurred at the higher sintering temperature when the matrix reacted with the reinforcement.

  1. To utilize the Williamson-Hall method to calculate the lattice strain, I believe we needs to confirm the linearity between sinθ and (Δ2θ)cosθ. How was the linearity? This information should be supplied.

Answers: The author did not confirm any linearity. The results of the lattice strain were automatically calculated by the XRD (PDXL: integrated x-ray powder diffraction software) and the results were obtained.

  1. In line 213, the authors have written that aggregates of TiB2particles are observed. However, in the XRD patterns, no TiB2is observed. Please provide explanation on this matter.

Answers: According to the investigation by Singh et al. (2019, https://doi.org/10.1007/s12613-019-1797-6), TiB2 was not formed in the XRD because the reaction between Ti and TiB2 was complete during the sintering at 1300 oC. This result is similar to our investigation. However, the authors have revised the sentence to agglomeration of in situ particles.

  1. In the friction experiment, considering that TiB2is more harder than the Ti matrix, I believe that usually the hard particles remain at the surface of the sliding track and reduces the contact area against the ball. However, I could not observe any TiB2particles in the sliding track of the ball in contrast to Fig.6. Please provide explanation on this matter.

Answers: The hard particles formed a bond with the matrix metal to improve the wear resistance of the composites. However, certain things happen when the ball slides on the metal surface;

If the ball is softer than the material, it can wear on the material’s surface. If it’s otherwise, there will be plowing of the material’s surface due to plastic deformation. In this study, the particles may be adhered to the material (compacted debris), as observed in the composites.

Reviewer 4 Report

The manuscript presents an interesting study of the mechanical characterization of the spark plasma sintering of titanium diboride-reinforced titanium matrix composites. However, the paper needs major revisions before it is processed further, some comments follow:

Introduction

The introduction section must be improved. In the introduction section, a comprehensive and exhaustive review of the state of the art in the field of the study must be provided. Please introduce and discuss more previous works, and highlight the experiments and results published previously.

Materials and methods

The subsection 2.1 must be divided into materials and morphological characterization.

Will be interesting to introduce a figure with the preparation process.

Results and discussion

Please remove the figures 5 and 6 from the tables.

Subsection 3.7.5. Please discuss every figure 16a, 16b, 16c....The same for figure 17.

Line 430. Please check the number of the Figure. Replace in all text.

Conclusions

The conclusion section must be improved. Add also quantitative results, recommendations and suggestions.  

References

There are too many self-citations. Please remove the unnecessary ones.

Author Response

The manuscript presents an interesting study of the mechanical characterization of the spark plasma sintering of titanium diboride-reinforced titanium matrix composites. However, the paper needs major revisions before it is processed further, some comments follow:

Introduction

The introduction section must be improved. In the introduction section, a comprehensive and exhaustive review of the state of the art in the field of the study must be provided. Please introduce and discuss more previous works, and highlight the experiments and results published previously.

Answers: This has been revised in the manuscript.

Materials and methods

The subsection 2.1 must be divided into materials and morphological characterization.

Will be interesting to introduce a figure with the preparation process.

Answers: This has been changed in the manuscript and a new figure has been included.

Results and discussion

Please remove the figures 5 and 6 from the tables.

Answers: This has been revised in the manuscript.

Subsection 3.7.5. Please discuss every figure 16a, 16b, 16c....The same for figure 17.

Answers: This has been revised in the manuscript.

Line 430. Please check the number of the Figure. Replace in all text.

Answers: This has been revised in the manuscript. Thank you.

Conclusions

The conclusion section must be improved. Add also quantitative results, recommendations and suggestions.  

Answers: This has been revised in the manuscript as ‘’ The titanium composites were consolidated via the SPS process at 1050 . The microhardness, relative densities, nanomechanical properties, tensile properties, microstructure evolution, and fractured surfaces were examined. In this study, adding TiB2 reinforcement to the titanium matrix improved the sintered specimens' relative densities from 97.5% to 99.3%. The Vickers microhardness values of the sintered sample increased from 188.1 HV1 to 304.8 HV1, about a 62% increment. The highest UTS value (514.3 MPa) was observed for the fabricated T2.5 composites before the UTS decreased with the volume of TiB2 reinforcement. The highest elongation value (27.1%) was observed for sintered T0.0 matrix, and a decrease in elongation values was noticed for the sintered composites. This proved that the tensile strength of the sintered composites increased at the expense of their elongations. The nanohardness improved from 6034 MPa to 9841 MPa, while the reduced elastic modulus improved from 156 GPa to 188 GPa upon adding TiB2 content. However, a slight drop was noticed for the T10 sample, possibly due to agglomeration. The result of the X-ray diffractograms showed the evolution of new phases, and there was no formation of the TiB2 phase in the sintered composites' XRD spectrum due to the incomplete reaction between the Ti and TiB2. The microstructures displayed lamellar structures with the distribution of whiskers, unreacted particles, and TiB corona. The sintered composites had better wear resistance than the T0.0 sample. The fractured surfaces of the sintered composites showed a combination of brittle and ductile behavior, while the sintered titanium showed ductile behavior.

References

There are too many self-citations. Please remove the unnecessary ones.

Answers: Self citations have been revised in the manuscript.

Reviewer 5 Report

Dear Authors. Although the article is very long, it is very interesting. My main complaint is the lack of appropriate proposals. Pay attention to their bullet points.

Author Response

#5

Dear Authors. Although the article is very long, it is very interesting. My main complaint is the lack of appropriate proposals. Pay attention to their bullet points.

Answers: This has been noted.

Round 2

Reviewer 1 Report

The revised paper can be accepted

Author Response

Thank you for your comment.

Reviewer 3 Report

The authors seemed to have tried to answer my questions and made some revisions; however, most are not correctly answered nor are reflected in the manuscript. The following points must be revised before it can be accepted.

1. The authors have answered “This present study was specifically carried out to assess titanium composites' mechanical and wear properties.” Indeed, this may be true. Therefore, I have pointed out, the present title is “Evaluation of the spark plasma sintering of titanium diboride-reinforced titanium matrix composites”. This means you are evaluating the effect of SPS method, not wear properties. This is evaluating “the wear property” of a material made by SPS and not the effect of SPS. Thus, the title should be something like “Evaluation of wear properties of titanium diboride-reinforced titanium matrix composites prepared by spark plasma sintering”. If my understanding is incorrect, please provide explanation.

2. The authors have answered “The authors repeated the images because the past manuscript showed the micrographs of two ceramics.“. But I want to say the a-Ti images are completely the same with your previous paper, as shown in Fig.R1 (Refer to attached file.). I do not think repeating an already published material is appropriate nor necessary. You should just refer to it using reference. What is the necessity of reusing the image? If it is necessary, please explain the reason.

3. The authors answered that “The difference in the relative density from T5.0 composite to T10.0 is very minute (about 0.1 range difference). The difference is not too significant, which is the reason for the observed error bar.” However, in the manuscript, you explain, “T10 sample, which showed a slight decline in relative density, attributable to the high amount of TiB2 reinforcement caused by the nucleation and development of the in-situ phase formed in the composite which influences the densification procedure” (p.5 line167). If this is not significant, is this discussion necessary? If not significant, I believe you should explain why the density saturated because it cannot say that it decreased.

4. The authors have answered that “The results of the lattice strain were automatically calculated by the XRD (PDXL: integrated x-ray powder diffraction software)”. In this case, how did you decide that you can used the Williamson-Hall method? Williamson-Hall method is not applicable to all materials. Is there a previous report in this system or a similar system? I think at least some kind of reason should be mentioned to say Williamson-Hall method is appropriate for this material. Please provide explanation.

Author Response

Comments and Suggestions for Authors

The authors seemed to have tried to answer my questions and made some revisions; however, most are not correctly answered nor are reflected in the manuscript. The following points must be revised before it can be accepted.

  1. The authors have answered “This present study was specifically carried out to assess titanium composites' mechanical and wear properties.” Indeed, this may be true. Therefore, I have pointed out, the present title is “Evaluation of the spark plasma sinteringof titanium diboride-reinforced titanium matrix composites”. This means you are evaluating the effect of SPS method, not wear properties. This is evaluating “the wear property” of a material made by SPS and not the effect of SPS. Thus, the title should be something like “Evaluation of wear properties of titanium diboride-reinforced titanium matrix composites prepared by spark plasma sintering”. If my understanding is incorrect, please provide explanation.

Answers: Thank you for your comment once again. Your suggestion for the title is okay, but I would like to add mechanical properties. Due to the fact mechanical properties were also carried out in this study. The new title would be ‘’ Evaluation of wear and mechanical properties of titanium diboride-reinforced titanium matrix composites prepared by spark plasma sintering.’’

  1. The authors have answered “The authors repeated the images because the past manuscript showed the micrographs of two ceramics.“. But I want to say the a-Ti images are completely the same with your previous paper, as shown in Fig.R1 (Refer to attached file.). I do not think repeating an already published material is appropriate nor necessary. You should just refer to it using reference. What is the necessity of reusing the image? If it is necessary, please explain the reason.

Answers: The image has been removed from the manuscript.

  1. The authors answered that “The difference in the relative density from T5.0 composite to T10.0 is very minute (about 0.1 range difference). The difference is not too significant, which is the reason for the observed error bar.” However, in the manuscript, you explain, “T10 sample, which showed a slight decline in relative density, attributable to the high amount of TiB2 reinforcement caused by the nucleation and development of the in-situ phase formed in the composite which influences the densification procedure” (p.5 line167). If this is not significant, is this discussion necessary? If not significant, I believe you should explain why the density saturated because it cannot say that it decreased.

Answers: From the previous answer to your question, the word ''not too significant'' was used to describe the occurrence in the error bar. However, a minute value range could still significantly impact during analysis, as observed in the relative density. The value (0.1 range difference) shows the effect of the high TiB2 within the Ti. 

  1. The authors have answered that “The results of the lattice strain were automatically calculated by the XRD (PDXL: integrated x-ray powder diffraction software)”. In this case, how did you decide that you can used the Williamson-Hall method? Williamson-Hall method is not applicable to all materials. Is there a previous report in this system or a similar system? I think at least some kind of reason should be mentioned to say Williamson-Hall method is appropriate for this material. Please provide explanation.

Answers: Thank you for your comment once again, it was reported in the data sheet that Williamson-Hall method was used for the lattice strain. The data sheet has been attached (page 10) for further verification.

Reviewer 4 Report

The authors addressed all my comments. In my opinion, the manuscript can be published in the present form.

Author Response

Thank you for your comment.